# Metal Accumulation and Biomass Production in Young Afforestations Established on Soil Contaminated by Heavy Metals

**DOI:** 10.3390/plants11040523

**Published:** 2022-02-15

**Authors:** Madeleine Silvia Günthardt-Goerg, Pierre Vollenweider, Rainer Schulin

**Affiliations:** 1Swiss Federal Institute for Forest, Snow and Landscape Research (WSL), Zürcherstrasse 111, 8903 Birmensdorf, Switzerland; pierre.vollenweider@wsl.ch; 2Institute of Terrestrial Ecosystems (ITES), ETH Zürich, 8092 Zürich, Switzerland; rainer.schulin@env.ethz.ch

**Keywords:** trace elements, metal extraction efficiency, phytoremediation, conifers, deciduous trees, understorey, forest ecosystem restoration

## Abstract

The restoration of forest ecosystems on metal-contaminated sites can be achieved whilst producing valuable plant biomass. Here, we investigated the metal accumulation and biomass production of young afforestations on contaminated plots by simulating brownfield site conditions. On 16 3-m^2^ plots, the 15 cm topsoil was experimentally contaminated with Zn/Cu/Pb/Cd = 2854/588/103/9.2 mg kg^−1^ using smelter filter dust, while 16 uncontaminated plots (Zn/Cu/Pb/Cd = 97/28/37/< 1) were used as controls. Both the calcareous (pH 7.4) and acidic (pH 4.2) subsoils remained uncontaminated. The afforestations consisted of groups of conifers, deciduous trees, and understorey plants. During the four years of cultivation, 2254/86/0.35/10 mg m^−2^ Zn/Cu/Pb/Cd were extracted from the contaminated soils and transferred to the aboveground parts of the plants (1279/72/0.06/5.5 mg m^−2^ in the controls). These extractions represented 3/2/3% of the soluble soil Zn/Cu/Cd fractions. The conifers showed 4–8 times lower root-to-shoot translocation of Cu and Zn than the deciduous trees. The contamination did not affect the biomass of the understorey plants and reduced that of the trees by 23% at most. Hence, we conclude that the afforestation of brown field sites with local tree species is an interesting option for their reclamation from an ecological as well as economic perspective.

## 1. Introduction

Soil contamination by metals can have either a lithogenic (parent material) or anthropogenic cause [1] and represents a global concern. In the present study, we focus on contamination by the heavy metals zinc (Zn), copper (Cu), lead (Pb) and cadmium (Cd). Soil pollution by metals is most severe in industrial areas and in regions with mining and metal-processing activities [2,3,4,5]. Most attention though has been given to metal contamination of agricultural land [6,7], which has been reviewed by [8]. In particular, Cd pollution of agricultural soil has been frequently studied [9,10]. However, wild plants [11,12] and forests [2,13] growing on contaminated soils can also pose environmental risks and threaten human health by deteriorating water quality and by contaminating metal transfer along the food chain [14].

Most soil contamination by heavy metals is due to atmospheric deposition or the land application of biowastes and agrochemicals and is thus concentrated in the topsoil, while the subsoil is mostly unaffected, due to the generally low mobility of trace elements [15,16,17]. Furthermore, the bioavailability of soil metals is generally limited, with site-specific differences related to variations in soil adsorption capacity, pH, texture and organic-matter content [18,19]. Cu and Pb show lower bioavailability than Zn and Cd [20,21]. Zinc and Cu are essential plant micronutrients and become toxic only at elevated concentrations [22,23,24], whereas Cd and Pb are not essential and have very low toxicity thresholds. Nevertheless, Cd is readily taken up by plant roots and translocated to shoots, due to its chemical similarity to Zn [25,26], whilst Pb is rather immobile and only sparingly internalized by root cells and translocated to plant shoots [27].

When the metal contamination of a soil exceeds threshold values, land-use restrictions—especially for food production—are generally applied, but plant production for non-food purposes is usually still desirable not just for economic but also for ecological reasons. A vegetation cover provides protection against the further dispersion of the contaminants by air and water and can greatly help in maintaining ecosystem services as far as possible. This strategy of managing metal-contaminated sites has considerable potential, as many examples of the spontaneous colonization of brownfield and mining sites by herbaceous [10,28] and woody [29] vegetation demonstrate. Revegetation with suitable plants showing little metal accumulation in the harvested parts can improve soil quality, reduce metal leaching into groundwater and allows for economically attractive biomass production [30,31,32,33,34,35,36,37]. As reviewed by [38], trials primarily with *Populus* and *Salix* tree species [39,40] or *Acer platanoides* seedlings [41] have shown an interesting potential for the restoration of metal-contaminated sites after spilling accidents [42] or the reclamation of brownfield sites [43,44]. Phytoextraction using high-yield crops [45] or trees (reviewed by [46]) has been suggested as a method to remediate soils, particularly those polluted with comparatively mobile metals.

Following up on a previous paper comparing increased metal concentrations among plant species in young forest vegetation planted on metal contaminated soil [47], here, the approach of re-using contaminated land is investigated from a phytoremediation perspective. The focus of this paper is on the transfer of contaminants from the soil into the roots and further into the aboveground parts of the planted forest vegetation, including understorey plants, coniferous and deciduous trees, on an area basis and dependent on the plant availability of the different metals. To account for species-specific variation in growth and the metal allocation of plants growing on metal-contaminated soils [48], we analysed the metal extraction capacity by plant groups. The hypothesis was that afforestation with the experimental vegetation used in this study is suited to manage metal-contaminated sites.

## 2. Results

Above- and below-ground plant Zn concentrations were increased in the contaminated soil (HM treatment) as compared to the controls (CO treatment). In many cases, plant Cu concentrations were also increased. Increased Pb concentrations were only found in roots. Increased plant Cd concentrations were found in roots, in understorey, deciduous tree foliage and in deciduous tree wood (Table 1). Wood and foliage differed in Zn, Cu and Cd concentrations of deciduous trees, but only in Cu of coniferous trees, and there were various significant differences among the groups (Table 1).

However, comparing metal concentrations between plants alone does not reflect their contributions to the cumulative mass of metal extraction over the experimental period, due to the very different biomass produced by the various plant species and groups. As in phytoremediation, it is the total amount of metal extracted from a given area over a given period of time that counts; therefore, we concentrate on the latter in the following. After four years of cultivation, on average 2254/86/0.35/10 mg m^−^^2^ Zn/Cu/Pb/Cd had been removed in total from the contaminated soil (HM treatment) and transferred to the aboveground plant biomass, versus 1279/72/0.06/5.5 mg m^−^^2^ in the CO treatment. Adding the amounts of metals accumulated in the roots, the total extraction of metals from the HM amounted on average to 2996/281/8.2/13.3 mg m^−^^2^ Zn/Cu/Pb/Cd versus 1391/88/2.3/6.1 mg m^−^^2^ when using the CO treatment. Thus, 2.2/3.2/3.6/2.2 times more Zn/Cu/Pb/Cd had been extracted by the entire plant biomass from the soil in the HM treatment than in the CO treatment (Figure 1, Table 2). There was a large variation in extraction between metals and plant groups, and a large variation in the allocation of the extracted metals in different plant parts. While Zn, which showed the highest accumulation among the four metals, was primarily allocated to foliage > wood in deciduous trees, Pb > Cu most commonly accumulated in the tree roots (Figure 1). Comparing metal amounts accumulated in different plant parts, the sequence was roots (R) < wood (W) < leaves (L) for Zn allocation in deciduous trees, L < W < R for Zn in conifers and for Cu in all trees, and R < L < W for Cd in deciduous tree organs (spruce at detection limit). More Zn and Cd was transferred into the aboveground parts by the deciduous trees than by the herbaceous plants in total, while less Cu was accumulated aboveground by the conifers than by the herbaceous plants. The deciduous trees, thus, showed a higher rate of total aboveground accumulation for the more mobile metals Zn and Cd than for the less mobile elements Cu and Pb, while the latter showed a higher rate of metal immobilization in conifer roots. As indicated by HM:subsoil interactions (Table 2), more Cd was extracted from plots with acidic soil than plots with calcareous subsoil, whereas metal extraction tended to be higher for the plots with calcareous subsoil in the control treatment (Zn and Cd in deciduous trees, Cu in spruce needles).

There were large differences between different plant groups in metal extraction efficiency and metal allocation with regard to the treatments (Figure 2). The trees showed lower shoot-to-root metal allocation ratios (SRMAR) in the HM than in the control treatment, especially for the conifers (spruce trees, Figure 2A). With mean values (averaged over both subsoil types) of 6.6 for Zn and 7.8 for Cd, the deciduous trees showed high SRMAR values for the relatively mobile metals Zn and Cd in the HM treatment, but for spruce the value for Zn was low (0.8), while it could not be determined for Cd (below detection limit in the aboveground biomass). In the control treatment, the SRMAR value of spruce for Zn and Cu was less than half of the group average SRMAR of the deciduous trees.

Relative to the total soil metal contents initially present, the soil-to-root metal transfer rates (SRMTR) of Zn/Cu/Pb/Cd reached 0.07/0.09/0.02/0.11 for the deciduous trees, and 0.12/0.13/0.03/0.11% for the spruce trees in the HM contaminated soil on average for both subsoil types (Figure 2B). Compared to the HM treatment, SRMTR values relating to total soil metals were higher for Zn, Cd and Cu, but similar for Pb in the control treatment. Soil-to-shoot metal transfer rates (SSMTR) were 0.5/0.05/0.8% for Zn/Cu/Cd in terms of uptake by the deciduous trees relative to the total amounts of these metals initially present in the contaminated soil (control treatment: 8.0% for Zn; Cu & Cd below detection limit, Figure 2C). Relative to the soluble soil metal fractions (Figure 2D), the SRMTR values increased in the HM treatment in the order (Zn = Cd < Cu < Pb) with spruce showing higher phytoextraction rates than deciduous trees (in the CO treatment Cd = Cu = Pb were below detection limit). When SSMTR was calculated relative to the soluble fractions of these metals (Figure 2E), uptake was 1.9/0.7/2.4% for Zn/Cu/Cd by the deciduous trees (control treatment: 613% for Zn; Cu & Cd below detection limit) and 0.4/0.3 for Zn/Cu uptake in spruce trees (CO: 137% for Zn). The roots therefore had initiated a solubilisation of Zn as a nutritional trace element in the controls. The understorey plants extracted 0.08/0.04/0.003/0.1% of the total soil Zn/Cu/Pb/Cd into their shoots (SSMTR, Figure 2C), which corresponds to 0.4/0.6/0.3/0.3% of the respective initially present soluble fractions (Figure 2E). Relative to the amounts of metals initially present in the soil (total or soluble), the soil-to-plant transfer of Zn was several times higher in the control than in the HM treatment for all plant groups (Figure 2B–E).

Separating the SSMTR (Figure 2C,E) into foliage and wood, the deciduous trees had transferred 0.3/0.02/0.4% of the total soil Zn/Cu/Cd into their foliage in the HM treatment, the spruce trees only 0.04/0.004% (Zn/Cu). Relative to the initially present soluble (“plant available”) soil metal fractions, these percentages amounted 1.2/0.5/0.3% Zn/Cu/Cd for the foliage of the deciduous, and to 0.2/0.05% for the foliage of the spruce trees. The percentages of total soil metals allocated to the aboveground wood amounted to 0.2/0.04/0.4% Zn/Cu/ Cd for the deciduous trees and to 0.06/0.02% Zn/Cu of the total soil metal contamination for the spruce trees. Relative to the initially present soluble soil metal fractions, the percentages amounted 0.7/0.5/0.3% Zn/Cu/Cd for the wood of the deciduous and to 0.2/0.05% for the wood of the spruce trees.

In total (foliage + wood + roots), soil metal extraction by the afforestations growing on the acidic subsoil amounted to 3.2/4.2/5.3/3.6% of the initially present soluble soil Zn/Cu/Pb/Cd fractions in the HM treatment, which for Zn, Cu and Pb was slightly less than observed for the calcareous subsoil (3.4/4.9/5.7/3.1).

The HM treatment did not affect the total biomass produced by the understorey plants on the acidic subsoil and even increased it (by 15%) on the calcareous subsoil. It led to a slightly reduced total tree biomass (deciduous trees on the acidic subsoil: −18%, on the calcareous subsoil −26%; spruce on the acidic subsoil: −20%; spruce on calcareous subsoil: +6.4%, with significant HM:soil interactions for spruce; Figure 3, Table 2).

## 3. Discussion

### 3.1. Biomass and Heavy Metal Stocks in Afforestations

Including roots, the total yearly biomass yield of the deciduous trees (7.5 t ha^−1^ year^−1^) was in the upper range of the values observed for poplars and willows in other studies, the total biomass production, including conifers and understory amounted 14 t ha^−1^ year^−1^ (Table 3). The group of spruce provenances planted in our study reached only (3.6 t ha^−1^ year^−1^), due to slow growth in the first two experimental years. Still, the total biomass yield of the afforestations remained below the 20 or 22 t ha^−1^ year^−1^ reported for agricultural crop plants such as *Zea mays* or *Sorghum bicolor* [49,50]. However, when comparing these data, it should be taken into consideration that the afforestations in our experiment do not represent a full rotation period. Apart from plant species and harvest age, factors such as planting techniques (seed, cutting, seedling age at planting), stand management and edaphic conditions also play an important role for optimizing woody plant production.

The soil contamination in the afforestations was in the range of values often observed on contaminated sites [51]. In spite of the very high concentrations of the contaminating soil metals in the HM treatment, the effect of the contamination on tree (*Populus/Salix/Betula*/*Picea*) growth was minor in our experiment, as compared to some pot trials with up to 50% decrease in the growth of *Acer* and *Tilia* [41]. Furthermore, the amount of metal taken up by the trees show large differences among various phytoremediation trials (Table 3). Metal extraction per square meter of soil was in the same order of magnitude as reported by [52] or [50], although the topsoil Zn/Cu/Cd concentrations differed largely (Table 3). With less severe soil contamination, higher Zn (and partly Cd) rates of extraction were observed [52,53,54,55] (Table 3).

With a duration of 4 years, our experiment only represents the initial phase of stand development in an afforestation. When long-living perennials such as trees are used for phytoremediation of metal-contaminated sites, the long-term evolution of metal stocks accumulated over time in these plants needs to be considered. In one study, large metal stocks were measured in 35-year-old *Pinus* trees growing alongside a highway, although the recent soil contamination was relatively low [53]. In another study, rather large metal stocks of high variability with a maximum in medium-aged trees (15 years) were found in a *Pinus massoniana* stand with low soil contamination [54]. However, considering other studies, no generalization is possible as of yet with respect to the long-term evolution of metal stocks in trees for contaminated sites (Table 3). According to the results obtained in the phytoremediation trials with *poplar* plantations [56], metal stocks in trees appear to primarily depend on the specific site and experimental conditions.

**Table 3 plants-11-00523-t003:** Comparison of dendroremediation results calculated from a range of references. L = Leaves, W = wood (shoot), R = roots, (-) = not analysed.

Topsoil Contamination (Total Extractable, mg kg^−1^ at Harvest)	Site	Species	Organs	Period (Years)	Extraction (mg m^−2^ year^−1^)	Yield (tha^−1^ year^−1^)	Reference
Zn/Cu/-/Cd = 1158/264/-/2.8 and 650/550/2	Caslano and Dornach Switzerland	*Salix viminalis*	LW	2 and 5	Zn/-/-/Cd = 330/-/-/0.1 and 155/-/-/0.1	5 or 6.6	[51]
Zn/Cu/-/Cd = 650/530/-/2	Dornach Switzerland	*Salix viminalis*	LWR	3 and 1	Zn/Cu/-/Cd = 128/6.3/-/1.3	4.3 and 14	[57]
Zn/Cu/Pb/Cd = 400/180/170/2.5	Copenhagen recycling center Denmark	*Salix viminalis*	LW	1	Zn/Cu/Pb/Cd = 35/0.8/0.04/10	0.9	[58]
Zn/Cu/Pb/Cd = 377/-/-/6.5	Campine region Belgium	*Salix viminalis* 8 clones	LW	4	Zn/Cu/Pb/Cd = 159/-/-/2	3.8	[49]
Zn/Cu/-/Cd = 174/81/-/1.3	Hradec Kralove Czech Republic	*Salix* and *Populus* clones	LWR	2	Zn/Cu/-/Cd = 1232/39/-/8.5	0.4	[52]
Zn/Cu/Pb/Cd = 295/24/283/2.8	Litavka River sediments Czech Republic	*Salix, Populus*	LWR	3	Zn/Cu/Pb/Cd = 100/-/2.3/2.7	12	[45]
Zn/Cu/Pb/Cd = 1563/112/-/16.7	Harbour Rotterdam, The Netherlands	*Populus* ‘Robusta’	W	33	Zn/Cu/Pb/Cd = 15/0.2/-/0.5	2	[56]
Zn/Cu/Pb/- = 56/17/27/-	Gao country, Sichuan, China	*Pinus massoniana*	LWR	stand 27	Zn/Cu/Pb/- = 3095/861/1453/-		[54]
Zn/Cu/Pb/Cd = 243/51/27/-	Erzurum Turkey	*Pinus sylvestris*	LWR	stand 35	Zn/Cu/Pb/- = 18′085/8′000/2′886/-		[53]
Zn/Cu/Pb/Cd = 972/173/1687/15	Chenzhou, China	*Populus deltoides*	LWR	5	Zn/Cu/Pb/Cd = 1410/38/148/61	10	[55]
Zn/Cu/Pb/Cd = 2854/588/103/9.2	Birmensdorf, Switzerland	Understorey + deciduous + conifer trees	LWR	4	Zn/Cu/Pb/Cd = 749/70/2/3	14	Figure 1 and Figure 3

### 3.2. Influence of the Uncontaminated Acidic and Calcareous Subsoil on Metal Uptake

In our study, we observed a lower total Zn and Cu uptake and higher total Cd uptake by the plants growing on the acidic subsoil than on the calcareous subsoil. With Zn concentrations being generally higher in the plants on the acidic subsoil, the lower amount of total Zn extraction was entirely due to the lesser production of biomass on the acidic subsoil, while for Cd the higher amount of extraction on the acidic subsoil was due to the fact that increased enrichment of this metal in plant tissues overcompensated for the reduction in biomass production. For Cu, the situation was more complex, as the subsoil effect on plant Cu concentrations differed by species and organ, but in any case the subsoil effect on biomass dominated the subsoil effects on plant Cu concentrations [47]. The higher biomass production on the calcareous subsoil can primarily be attributed to the observed enhanced root growth of the deciduous trees in the nutrient-rich calcareous subsoil, as reported earlier [59]. Using root cores, the latter study found a root density at 50–75 cm depth of 0.2 mg m^−3^ in the calcareous subsoil, compared to 0.1 mg m^−3^ in the acidic subsoil in both the HM and the CO treatment. 

### 3.3. Metal Allocation Ratios and Soil-to-Plant Metal Transfer

While the shoot-to-root metal allocation ratios (SRMAR), calculated in the present study as ratios between total amounts of metal allocation (g g^−1^) in shoot and root biomass, are not directly comparable to the transfer factors (TF) calculated as ratios between shoot and root metal concentrations in other studies, the differences in SRMAR values among the various metals obtained in our study were similar to those among TF values for these metals found in studies with medicinal plants [60] and wild herbal plants [28,61]. Comparable to our results for spruce, low root-to-shoot translocation (TF below 1) has been reported for shrub [62] and herbaceous species [45,63] at total soil metal concentrations that were lower than in our afforestations. This was also observed in the case of higher soil contamination for several herbaceous and tree species [64]. The SRMAR values we obtained for the trees growing on the uncontaminated soil appear to be high, but we found no values in other studies for uncontaminated soil for comparison. Low percentages of root-to-shoot metal translocation on highly contaminated soils may be partially due to high degrees of metal immobilization in the soil and in the root tissues and the effective filtering of excessive concentrations of metals from the soil solution by the root endodermis barrier.

The SRMAR values were smaller in all trees for Cu than for Zn. The SRMAR values found for the deciduous tree group (Zn/Cu 6.6/0.7) were larger than those for spruce. While root-to-shoot translocation of Cd was not determined in the conifer tree group due to detection limits, the SRMAR values found for Cd in the deciduous tree group were the highest (7.8) recorded for all target metals in the contaminated soil of our study. In line with our results, the available evidence shows a negative correlation between soil Cd concentration and root-to-shoot translocation of Cd. This may relate to Cd toxicity, considering the low filtering efficiency by the root endodermis control barrier for this toxic element.

Additionally, the relative soil-to-root (SRMTR) and soil-to-shoot metal transfer rates (SSMTR) calculated in the present study are not directly comparable with biological concentration factors (BCF) and biological accumulation factors (BAF), used in other studies, as the latter refer to concentration ratios similar to the TF and not to ratios between amounts of metals in soil and plant metal pools. However, the fact that the SRMTR in the two tree groups of our study never exceeded 1% for any of the four target metals is in line with the low accumulation rates (BCF) reported for these metals in other investigations [11,55,61,65], with few exceptions for Zn [41,66]. For Zn and Cd, the variations in SRMAR showed similar patterns, indicating similar mobility of these metals within plants. Similarly, the low SSMTR values found in our study (<1%) are in agreement with the generally low BAF values reported in the literature [28]. The fact that the relative rates of metal transfer from the soil into below- and aboveground plant parts were higher for the uncontaminated soil than for the contaminated soil agrees with the observation that low soil Zn and Cu concentrations are usually associated with high BCF and BAF values [11,67,68,69].

### 3.4. Soil Metal Solubility in Relation to Plant Uptake and Allocation

The magnitude of the soluble soil metal fractions relative to the total metal contents in the contaminated topsoil (Zn 24%, Cd 33%, Cu 7% and Pb 1%) were in the range of the values reported for the four target metals in other studies [45,70]. According to the literature, the relative magnitude of these fractions shows little dependency on the vegetation [50,71]. As it is a wide-spread notion [72,73,74] that soluble soil metal concentrations are better predictors of soil metal uptake by plants than total soil metal concentrations, we determined soil-to-plant transfer rates also relative to the magnitude of soluble soil metal pools. Given that by definition the latter pools are only a fraction of the respective total soil pools, SRMTR and SSMTR values based on the soluble pools are always higher than those based on total soil metal pools. For Zn, the relative extraction from the soluble soil metal pool was up to 76 times higher than from the total soil metal pool in the control treatment and up to four times higher in the HM treatment in the case of the deciduous tree group. For Cu and Cd, transfer rates relating to the soluble metal pools were up to 14 (Cu) and three (Cd) times higher than for the respective total soil metal stocks in the HM treatment. The largest ratio between transfer rates relating to soluble and total soil metal stocks was found for Pb with a value of 103, reflecting the low solubility of Pb in soil.

The fact that in the control treatment the amounts of Zn transferred into the aboveground biomass, in particular that of the deciduous trees, greatly exceeded the initial pools of soluble soil Zn on both subsoils indicates the capacity of plants to substantially enhance Zn availability in soils [75].

### 3.5. Accumulated Soil Metal Stocks in Foliage and Aboveground Wood

The extraction of the metal contaminants from the soil and their allocation in the woody plant parts showed considerable variation among metals and plants. Only a low share of the accumulated Zn had been allocated in deciduous tree wood (0.2%), and an even lower share was observed in the spruce wood (0.06%), despite the much larger biomass produced in form of aboveground wood than in form of foliage. The rates of Cu were even less (0.04% for deciduous and 0.02 for conifer trees). Metal stocks accumulated in the foliage of deciduous trees are recycled annually to the ground with leaf litter fall, whereas metals accumulated in woody tissues represent a much more stable aboveground plant metal stock. In the literature, metal concentrations rather than stocks are usually reported, but there are a few studies in which plant metal stocks were reported, allowing for a direct comparison with the results of our study (Table 3). In particular, the Zn and Cd stocks found in the wood of species such as *Populus, Salix* or *Betula* grown on soil contaminated with >1000 mg/kg Zn and 15-120 mg/kg Cd generally represented 8–40% of the respective stocks in the foliage [47,55,58,68,76]. While most stem biomass consists of wood tissues, metals are predominantly accumulated in bark tissues. The accumulation of Zn/Cu/Cd in bark can reach 46–94% of the metal stocks in the wood compartment [56,76]. Stem Pb can even accumulate up to 99% within the bark, depending on the trunk radius of the tree [77]. Taking appropriate precautions, wood without bark may thus be produced for commercial use even on contaminated soil.

### 3.6. Phytoremediation Potential of Afforestations for Metal-Contaminated Soils

Even without a replenishment of the soluble soil metal fraction and without a concentration-dependent decrease in the rate of metal uptake over time, it would take more than 350 years to reduce the soluble soil Zn concentrations below critical limits in the contaminated soil [78]. For Cu, approximately 30 years would still be needed. Only the soluble Cd concentration should have reached a tolerable level within slightly more than a year. However, as our results for Zn uptake from the uncontaminated control soil demonstrate, the replenishment of soluble soil metal pools from the pools of less soluble soil metals cannot be ignored and may be considerable over time scales of years or less. However, even if the above assumptions were valid, the time required to remove just the excess of soluble metals would be prohibitive in practice for Cu. Trying to bring down the total concentrations of the contaminating metals to tolerable levels with our afforestations would be even more challenging.

The extraction rates obtained in our experiment are consistent with those of other trials in which willow and poplar plantations with short-rotation coppicing were used to extract contaminating soil metals. For example, it was concluded in one study that 55–108 years would be needed using the selected clones of *Salix viminalis* or 247–805 years with poplar species, so as to lower the contamination of a soil with 5.7 mg kg^−1^ Cd to a tolerable concentration of 2 mg kg^−1^ Cd [79]. While we did not find the afforestation of our study to be a realistic option to clean soil with a metal-contamination as in our HM treatment, our results show that afforestation can still be useful in reclaiming soils for plant production, while stabilizing the soil against erosion and associated contaminant dispersal and at the same time reducing the contamination in the very long run. In our study, the HM treatment had only a limited impact on wood production. Scaling-up the results from our plots to entire stands, the average annual wood production would have been 4 t ha^−1^ for the deciduous trees and 1.6 t ha^−1^ year^−1^ for spruce in the HM treatment. This is comparable to findings for pine species growing on uncontaminated soil [43]. Considering the thick resprouting of the poplars and willows we observed after the yearly coppicing, these trees seem to be particularly suited for renewable-energy production, even on sites heavily contaminated with metals [80]. Additionally, short-rotation plantations with these species can still help to develop an interesting biodiversity [40]. Promising forestry-based phytoremediation trials include a mixed woodland plantation in which the contamination of a soil with 180/60/200/1 mg kg^−1^ Zn/Cu/Pb/Cd was reduced within 14 years by 47/48/44/52%, respectively, while leaving that soil under grassland vegetation resulted in much less contaminant removal [81]. In addition, compared to afforestation, grassland use would have the disadvantage of a much larger risk of contaminant transfer into the human food chain.

Concluding that great ecological as well as economic benefits can result from employing afforestation as a strategy to reclaim and manage metal-contaminated soils, it should be noted that substantial contributions to the success of this strategy can come also from understorey plants. Native herbs and grasses from various plant families have shown appreciable metal tolerance and phytostabilization capacities and thus demonstrated suitability for their use in the reclamation of brownfield sites with mixed soil metal contamination [10,12,64,82]; see also the review by [83]. Species of particular interest include *Trifolium repens*, *Lolium perenne* [65], *Stipa tenacissima*, *Artemisia herba-alba* [28], *Gentiana pennellina* [67], *Inula viscosa* [11], *Sinapis arvensis, Silybum marianum* [61], *Achillea millefolium, Ranunculus ficaris* [60], *Baccharis latifolia, Lepidium bipinnatifidum* [66] or *Armerietum halleri* [84]. While there is generally little economic interest in native understorey plants (apart from the occasional medicinal use of some understorey plants [60]), and although non-harvested plants do not contribute to the long-term reduction of soil metal stocks, they can still make important contributions to regreening, soil stabilisation, the prevention of contaminant dispersal by erosion or leaching into groundwater and an improvement of soil properties, and thus to the restoration of environmental services of the site. However, the potential problem of metal transfer into the nutrient chain remains and needs to be taken care of [60].

In summary, our results show that afforestation using native plants can be a suitable strategy to manage even heavily metal-contaminated brownfield sites. The attractivity of this strategy further gains if such afforestations are targeting novel products, such as bio-plastics and bio-chemicals, and if they contribute [85] to the development of sustainable bio- and wood-economy alternatives (reviewed by [86]), in compliance, e.g., with European incentives [85]. Finally, afforestations using native trees and understorey plants adapted to local site conditions do not only represent an attractive option for re-claiming metal-contaminated brownfield sites in an aesthetically and socially acceptable way, combining the restoration of ecosystem services with the production of valuable plant biomass, but at the same time they may also contribute to achieving a less carbonated economy in addressing issues related to climate change.

## 4. Materials and Methods

The experiment was carried out 1999–2003 on 32 plots of 3 m^2^ surface area each with constructed soil profiles. The 1.5 m deep subsoils, a calcareous sandy loam (pH 7.4) and an acidic loamy sand (pH 4.2), remained uncontaminated. The texture of the calcareous subsoil was 74:16:10 and that of the acidic subsoil 87:8:5 sand:silt:clay (fractions in %). The organic carbon content (C_org_) was 21 g kg^−1^ in the calcareous and <1 in the acidic subsoil. The carbonate-free 15 cm topsoil of all plots had a texture of 36:49:15 sand:silt:clay, a C_org_ < 1 g kg^−^^1^ and a pH of 6.55. The topsoil of eight plots with calcareous and eight plots with acidic subsoil was experimentally contaminated with metal smelter filter dust (Zn/Cu/Pb/Cd = 800/170/15.5/0.27 g kg^−^^1^ [15] and some additional Cd oxide). In this way, the mean Zn/Cu/Pb/Cd concentrations were increased to 2854 ± 872/588 ± 206/103 ± 27/9.2 ± 3.6 mg kg^−^^1^ (HM treatment), as compared to 97 ± 1/28 ± 4/37 ± 3/< 0.1 mg kg^−^^1^ in the uncontaminated topsoil of the 16 control plots (CO treatment). The total amounts of Zn/Cu/Pb/Cd extractable with 2M HNO_3_ were 360/74/13/1 g m^−^^2^ in the HM treatment and 12/3.5/4.7/< 0.1 g m^−^^2^ in the CO treatment. The soluble (or “mobile”) fractions extractable with 1M NH_4_NO_3_ (better suited for estimating soil metal bioavailability than complexing agents [72,73]), which were considered as plant available) amounted to 86/5.4/0.1/0.4 g m^−^^2^ in the HM treatment versus 0.16/< 0.1 < 0.1 < 0.1 g m^−^^2^ in the control treatment. All plots were planted with identical groups of plants, with full plant-position randomization in each group:-The understorey group (u) consisted of four specimens of tansy (*Tanacetum vulgare* L.) grown from root cuttings, four small sedge plants of (*Carex sylvatica* Hudson), one specimen of ransom (*Allium ursinum* L.) grown from bulb, plus three oak (*Quercus pubescens* Wild.), three beech (*Fagus sylvatica* L.) and three spruce seedlings grown from seeds directly sown in the plots.-The deciduous tree group (d) consisted of two birch (*Betula pendula* Roth), two willow (*Salix viminalis* L.) and four poplar (*Populus tremula* L.) trees grown from cuttings.-The conifer group (s) consisted of six spruce trees (*P. abies* (L.) Karst., six provenances from 500 to 1800 m a.s.l.) grown from three-year-old nursery seedlings.

For more information on the experimental system, readers are referred to our previous paper [47]. The experiment lasted four years. Willows and poplars were coppiced annually, and the aboveground biomass of the herbaceous understorey plants was also harvested each year. At the end of the experiment, all plants were harvested individually. The dry mass of the trees was determined separately per organ (foliage/wood/roots). The roots and shoots were difficult to separate in the case of *Carex and Allium*, and the roots of *Tanacetum* were not associable to individual plants; therefore, they were classed as “non-associated roots”. Root residues that remained in the ground were collected by sieving the excavated soil after the final harvest (= non-associated root fraction). The root material was thoroughly washed to remove any adhering soil particles. Plant (organ) samples were ground to a fine powder (Retsch MM2000 zircon oxide-bowl ultra-centrifuge mill), digested in a high-pressure microwave system (UltraClav by Milstone: 240 °C, 12 MPa) and analysed in duplicate (range < 10%) by means of ICP-OES (Optima 7300DV by Perkin Elmer) at the WSL central laboratory, according to ISO 17025.

Calculations and statistics: The amount of metals accumulated by the experimental plants was determined by multiplying the biomass of plant parts with their respective metal concentrations (excluding values below the detection limit dl, <0.1 mg kg^−1^) and summing up the products obtained for each harvest per plant group and plot area over the entire experimental period. For non-associated roots, we used the weighted mean metal concentrations of the tree roots. To calculate metal amounts in the topsoil per m^2^ plot area (total or soluble, respectively), the metal concentrations of the topsoil were multiplied by the topsoil dry mass per m^2^ plot area (126 kg m^−2^ = 0.84 g m^−3^ topsoil × 0.15 m topsoil depth). Metal-specific shoot:root metal allocation ratios (SRMAR) were calculated as ratios between the respective masses of shoot and root metal (g g^−1^) accumulated per plot by a respective plant group. Similarly, metal-specific relative soil-to-root metal transfer rates (S**R**MTR) and the relative soil-to-shoot metal transfer rates (S**S**MTR) were calculated as the ratios between the masses of a metal in the plant part and in the soil, respectively. The transferred metal masses were related either to the total or to the soluble soil metal mass, as indicated in the specific context. Given the two subsoil types and two levels of topsoil HM contamination, there were four treatment combinations, each replicated in eight plots. After log-transformation of the data, the effects of metal contamination, subsoil type and their interaction were tested by means of Type III ANOVA, fitting a general linear model (GLM) for each dependent variable, followed by post hoc pairwise Tukey‘s studentized range (HSD) test, to test differences between individual means. Non-transformed data were used for plotting graphs within figures. All statistical analyses were performed using the SAS release 9.4 program package (SAS Institute Inc. Cary, NC, USA).

## Figures and Tables

**Figure 1 plants-11-00523-f001:**
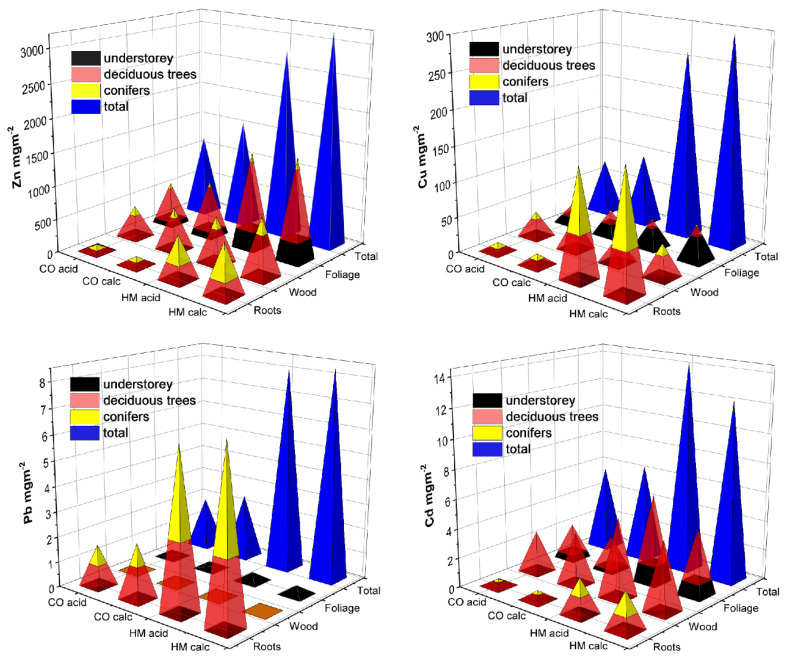
Mean (*N* = 8) rates of soil-to-plant metal transfer (mg metal per m^2^ ground area) over 4 years after establishing afforestations by plant groups (understorey plants, deciduous trees, conifers) and plant parts (roots, aboveground wood, foliage). Pb was below detection limit in tree wood (brown squares) and above detection limit only in understorey shoots from the HM treatment. Cd was below detection limit in conifer wood and foliage. CO = uncontaminated control plots, HM = plots with heavy metal contaminated topsoil, calc = calcareous subsoil, acid = acidic subsoil; total = all parts of all plants + non-associated roots collected by sieving the soil after the final harvest. ANOVA see Table 2.

**Figure 2 plants-11-00523-f002:**
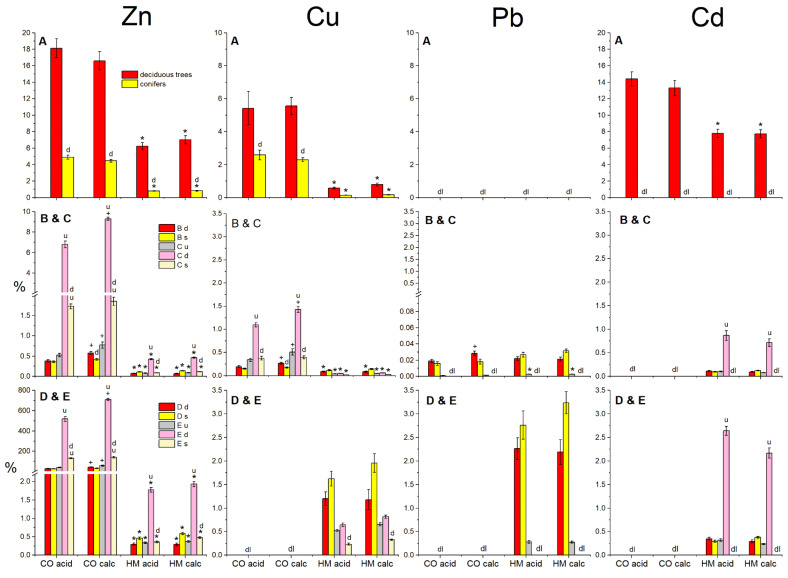
Soil metal contamination (HM = contaminated topsoil vs. CO = no contamination) and subsoil (calc = calcareous vs. acid = acidic) effects on (**A**) the ratios (g g^−1^) between metal amounts allocated to shoots and roots (SRMAR = shoot-to-root metal allocation ratio) and (**B**–**E**) on the metal amounts transferred into the roots (**B**,**D**) and shoots (**C**,**E**) of the experimental plants relative (in % of mass) to the amounts of contaminating metals in the soil (SRMTR = relative soil-to-root metal transfer rate; SSMTR = relative soil-to-shoot metal transfer rate). The rates of metal transfer into roots and shoots are given relative to the total amount of the respective soil metal (**B**,**C**), as well as relative to the magnitude of its soluble pool in the soil (**D**,**E**). Bars represent mean values ± SE (N = 8) per metal treatment and subsoil by plant groups (u = understorey plants, d = deciduous trees, s = conifers, i.e., spruce). No values could be determined for the SRMAR and SRMTR of the understorey plants, as their roots could not be clearly separated, and where metal concentrations were below detection limit. Asterisks (*) indicate significant differences (*p* < 0.05) between HM and CO treatment according to Tukey test; a plus (+) indicates a significant difference between the two subsoil types; the letters u, d, s above a bar denote significant differences of the respective plant group to the plant group indicated by the letter within a treatment.

**Figure 3 plants-11-00523-f003:**
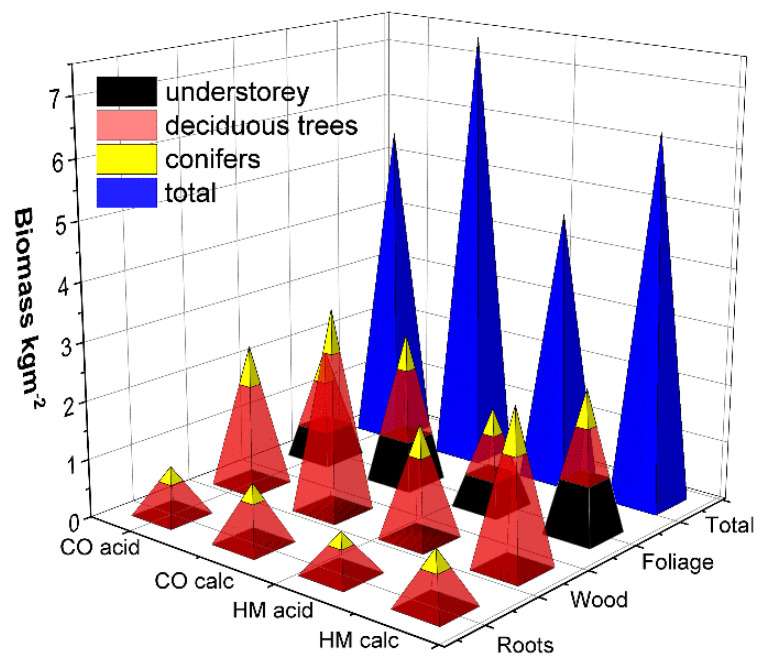
Mean biomass (*N* = 8) of the afforestations at the final harvest 4 years after plantation, by plant parts and plant groups. ANOVA see Table 2, notation as in Figure 1.

**Table 1 plants-11-00523-t001:** Mean ± SE (N = 8) metal concentrations per organ for each plant group and treatment at the end of the experimental period. Groups: u = understorey plants (*Allium, Tanacetum, Carex,* seedlings of *Quercus, Fagus* and *Picea*). d = deciduous trees (*Betula, Populus, Salix*). s = coniferous trees, *Picea abies*; R = roots, W = wood, L = leaves (foliage). nd = not determined due to metal concentrations below detection limit (dl = 0.1 mg kg^−1^). Asterisks (*) indicate significant differences (*p* < 0.05) between HM and CO treatment according to Tukey test; a plus (+) indicates a significant difference between the two subsoil types each in the HM or CO treatment; the letters u, d, s denote significant differences of the respective plant group to the plant group indicated by the letter within a treatment; R, W, L denote significant differences of the respective plant organ within a group and treatment.

	Zn				Cu				Pb				Cd			
	HM		CO		HM		CO		HM		CO		HM		CO	
	Acid	Calc	Acid	Calc	Acid	Calc	Acid	Calc	Acid	Calc	Acid	Calc	Acid	Calc	Acid	Calc
u R	1544 ± 61.2 ds	1432 ± 53.5 ds	109.7 ± 14.4 *ds	103.0 ± 13.9 *ds	579.4 ± 56.5 ds	505.3 ± 23.8 ds	31.9 ± 10.5 *	30.7 ± 7.7 *	10.9 ± 0.7 ds	10.4 ± 0.7 d	2.0 ± 0.09 *s	2.1 ± 0.16 *	4.8 ± 0.19 d	4.6 ± 0.26 d	0.4 ± 0.03 *d	0.4 ± 0.03 *d
u W	147.2 ± 9.5 ds	132.5 ± 13.2 ds	65.6 ± 6.6 *ds	76.5 ± 12.8 *s	16.8 ± 0.6 ds	22.6 ± 3.2 +ds	14.6 ± 0.4 *ds	13.4 ± 0.6 *ds	nd	nd	nd	nd	nd	nd	nd	nd
u L	253.2 ± 13.5 Wds	187.1 ± 9.4 +Wd	76.2 ± 3.2 *d	74.3 ± 1.9 *ds	16.5 ± 1.0 ds	17.5 ± 1.4 ds	14.5 ± 1.8 s	12.8 ± 0.7 *ds	3.7 ± 0.7	3.0 ± 0.6	nd	nd	1.2 ± 0.06 d	0.7 ± 0.06 +d	0.5 ± 0.03 *d	0.3 ± 0.02 *+d
d R	876.2 ± 59.4 us	598.4 ± 40.2 +us	158.8 ± 5.0 *u	141.6 ± 3.6 *+u	181.9 ± 11.9 us	131.7 ± 11.9 +us	16.7 ± 0.4	16.5 ± 0.7 *	7.7 ± 0.4 us	6.0 ± 0.3 +us	2.5 ± 0.15 *	2.3 ± 0.16 *	3.3 ± 0.16 u	1.9 ± 0.01 +us	1.0 ± 0.04 *us	0.7 ± 0.02 *+us
d W	257.4 ± 5.0 us	171.6 ± 5.9 +us	127.5 ± 5.1 *u	103.4 ± 4.1 *+	7.1 ± 0.2 us	6.1 ± 0.1 +us	6.4 ± 0.2 *us	6.0 ± 0.2 us	nd	nd	nd	nd	1.5 ± 0.04	0.8 ± 0.06 +	0.8 ± 0.06 *	0.5 ± 0.02 *+
d L	1270 ± 44.8 Wus	905.7 ± 29.5 +Wus	560 ± 13.6 *Wus	424.5 ± 11.1 *+Wus	11.2 ± 02 Wus	10.6 ± 0.4 Wus	10.8 ± 0.2 Ws	10.2 ± 0.2 Wus	nd	nd	nd	nd	3.8 ± 0.17 Wu	2.0 ± 0.10 Wu	2.1 ± 0.10 Wu	1.2 ± 0.07 Wu
s R	1731 ± 88.4 du	1874 ± 96.7 du	164.1 ± 4.3 *u	157.7 ± 5.7 *u	365.5 ± 24.0 du	322.7 ± 32.4 du	19.7 ± 0.7 *	18.4 ± 0.8 *	14.6 ± 1.0 du	12.4 ± 1.0 d	2.9 ± 0.33 *u	2.3 ± 0.26 *	5.0 ± 0.25 d	4.6 ± 0.27 d	0.5 ± 0.03 *d	0.5 ± 0.06 *d
s W	178.5 ± 9.2 du	180.7 ± 9.5 du	127.7 ± 4.7 *u	122.3 ± 10.4 *u	9.6 ± 0.5 du	11.1 ± 0.8 du	9.0 ± 0.4 du	9.3 ± 0.5 du	nd	nd	nd	nd	nd	nd	nd	nd
s L	185.8 ± 8.3 du	191.3 ± 5.9 d	101.6 ± 3.9 *Wd	108.9 ± 2.9 *du	3.60 ± 0.3 Wdu	4.5 ± 0.2 Wdu	3.5 ± 0.3 Wdu	4.3 ± 0.1 Wdu	nd	nd	nd	nd	nd	nd	nd	nd

**Table 2 plants-11-00523-t002:** ANOVA results for the data presented in Figure 1 and Figure 3: Error probabilities of significant effects by plant parts and plant groups on amounts of plant metal accumulation for the HM treatment (heavy metal contamination of topsoil vs. no contamination), the subsoil type (calcareous vs. acidic) and their interaction (HM:soil). u = understorey plants, d = deciduous trees, s = coniferous spruce trees; R = roots, W = wood, L = leaves (foliage), total = R + W + L + non-associated roots; ns = not significant (*p* > 0.05), nd = not determined due to metal concentrations below detection limit.

	Zn		Cu		Pb		Cd		Biomass	
Group/ Organ	HM	Subsoil	HM	Subsoil	HM	Subsoil	HM	Subsoil	HM	Subsoil
u	<0.0001	0.0083	<0.0001	0.0011	<0.0001	ns	<0.0001	0.0002	ns	<0.0001
d R	<0.0001	0.0399 HM: soil 0.0189	<0.0001	ns	<0.0001	ns	<0.0001	HM: soil 0.0309	0.0011	<0.0001
d W	<0.0001	<0.0001	ns	<0.0001	nd		<0.0001	HM: soil 0.0024	<0.0001	<0.0001
d L	<0.0001	0.0030 HM: soil 0.0068	0.0046	0.0003	nd		<0.0001		0.0001	<0.0001
s R	<0.0001	0.0042	<0.0001	ns	<0.0001	ns	<0.0001	0.002	ns	<0.0001 HM: soil 0.0289
s W	<0.0001	0.0077	ns	0.0088	nd		nd		ns	<0.0001 HM: soil 0.0053
s L	<0.0001	0.0035	ns	<0.0001 HM: soil 0.0357	nd		nd		0.0072	<0.0001 HM: soil 0.0004
**total**	<0.0001	<0.0001 HM: soil 0.028	<0.0001	0.0018	<0.0001	ns	<0.0001	0.0022	<0.0001	<0.0001

## Data Availability

Data are available upon request by contact with the corresponding author.

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
