# Peer review of "Metal Accumulation and Biomass Production in Young Afforestations Established on Soil Contaminated by Heavy Metals"

_plants, 2022, doi:10.3390/plants11040523_

Round 1
Reviewer 1 Report
The first sentence of the Abstract ‘Restoration of ecosystem services on metal-contaminated sites can be achieved whilst producing valuable plant biomass’ does not appear to reflect any statements made in the Introduction or any evidence presented from earlier studies. I see nothing about ecosystem services.
As a biologist, I take exception to the use of the term ‘adapted plants” on Line 59. This infers some kind of genetic selection, whereas I think the authors mean ‘suitable plants’
A little more information in the Methods section would be helpful, rather than just referring the reader to the 2019 lysimeter paper in Plant and Soil. The 15 cm topsoil layer had a pH 6.55. I couldn’t find SOM content in either paper. A little more in this paper on the constructed soil profiles would be more useful.
Listing of the metal concentrations in the filter dust from a non-ferrous metal smelter would be helpful.
I would have liked to see a small table of metal concentrations in plants, rather than just the biomass and mass balance calculations.
In neither paper could I find detail of rooting of the perennial species into the subsoil. This would have been very valuable. The 23% reduction in growth of tree species probably reflects this.
Pointed histogram bars in Figures may be suitable for a presentation but I don’t think they are ideal for print, but it’s not my paper.
Author Response
Please see the attachment."

Reviewer 2 Report
Review Remarks on MS: entitled: ‘Above- and belowground metal accumulation and biomass production in young afforestations established on soil contaminated by heavy metals’.
The followings are some general points and scientific queries needed to be addressed in the manuscript:
- Please mention the years in which the experiment was performed in the materials and methods section.
- The style of citing references at certain places needs to be done according to the journal’s guidelines. For example, As reviewed by [38], [Line 62].
- Please check line 16.
- Please clearly state the hypothesis of the study in the introduction.
- It is recommended to include a schematic representation for the mechanism of reclamation of heavy metals in brownfields.
- There are some grammatical issues, please pay attention.
- Please limit or avoid the use of personal pronouns.
- In Table 2, what does this sign - depict the absence or please state it.
Thus, overall, the presentation is average.
Reviewer 3 Report
Dear Editors and Authors,
I have read your manuscript carefully. In my opinion, it is a very interesting paper. Afforestation is a crucial part of phytoremediation polluted areas. The manuscript confirms the great potential of using dendroremediation in practice. Below I am presenting my comments:
- I suggest shortening the title since the work describes both underground and aboveground parts, it can be omitted in the title because the entire plant is described;
- please, use different keywords than words in the title;
- phytoremediation with the use of trees is dendroremediation; please consider adding this definition in the Introduction section to be more precise (you can cite https://doi.org/10.1007/978-3-319-99651-6_12 and https://doi.org/10.1007/978-1-4020-4999-4_3
- please add information about investigated tree species at the end of Introduction section;
- I really appreciate the ingenuity in presenting the results, but the data in Figures 1 and 3 are very difficult to read. Could you use a simpler graph and plot the statistics from Table 1 on it as well?
- Research on the use of trees in remediation is currently carried out by many scientists around the world, and the results of these studies are increasingly published in high-scoring journals. It turns out that tree species often show completely different features and possibilities than other plants, especially those of small size. In this regard, I believe that only tree species studies should be cited in Table 2 in order to make sense in the peer-reviewed manuscript.
- Please check figure 2 if everything is okay graphically. Some bars are framed, others are not.
- Does the description of the results in the Results section include materiality or non-significance for the data obtained?
- Discussion of the results must focus on comparison with other studies on woody species. It is very important.
- "the 15 cm topsoil was experimentally contaminated with smelter filter dust" - How is it ensured that the added pollutants do not penetrate into the deeper layers of the soil and further into the groundwater? Was the root system of the studied trees limited to 15 cm?
- Please indicate clearly what are the differences between the research in the manuscript published in Plant and Soil and the reviewed paper. Please indicate the novelty statement of each of these papers.
Best regards,
Reviewer
Round 2
Reviewer 3 Report
Accept in present form.